# Identification of Inflammatory Proteomics Networks of Toll-like Receptor 4 through Immunoprecipitation-Based Chemical Cross-Linking Proteomics

**DOI:** 10.3390/proteomes10030031

**Published:** 2022-09-01

**Authors:** A. D. A. Shahinuzzaman, Abu Hena Mostafa Kamal, Jayanta K. Chakrabarty, Aurchie Rahman, Saiful M. Chowdhury

**Affiliations:** 1Department of Chemistry and Biochemistry, The University of Texas at Arlington, Arlington, TX 76019, USA; 2Pharmaceutical Sciences Research Division, Bangladesh Council of Scientific and Industrial Research (BCSIR), Dhaka 1205, Bangladesh; 3Advanced Technology Cores, Dan L Duncan Comprehensive Cancer Center, Baylor College of Medicine, Houston, TX 77030, USA; 4Quantitative Proteomics and Metabolomics Center, Columbia University, New York, NY 10027, USA

**Keywords:** Toll-like receptor 4, human embryonic kidney, protein interactions, proteomics, mass spectrometry, macrophage myristoylated alanine-rich C kinase substrate, lipopolysaccharide, statin, cross-linker

## Abstract

Toll-like receptor 4 (TLR4) is a receptor on an immune cell that can recognize the invasion of bacteria through their attachment with bacterial lipopolysaccharides (LPS). Hence, LPS is a pro-immune response stimulus. On the other hand, statins are lipid-lowering drugs and can also lower immune cell responses. We used human embryonic kidney (HEK 293) cells engineered to express HA-tagged TLR-4 upon treatment with LPS, statin, and both statin and LPS to understand the effect of pro- and anti-inflammatory responses. We performed a monoclonal antibody (mAb) directed co-immunoprecipitation (CO-IP) of HA-tagged TLR4 and its interacting proteins in the HEK 293 extracted proteins. We utilized an ETD cleavable chemical cross-linker to capture weak and transient interactions with TLR4 protein. We tryptic digested immunoprecipitated and cross-linked proteins on beads, followed by liquid chromatography–mass spectrometry (LC-MS/MS) analysis of the peptides. Thus, we utilized the label-free quantitation technique to measure the relative expression of proteins between treated and untreated samples. We identified 712 proteins across treated and untreated samples and performed protein network analysis using Ingenuity Pathway Analysis (IPA) software to reveal their protein networks. After filtering and evaluating protein expression, we identified macrophage myristoylated alanine-rich C kinase substrate (MARCKSL1) and creatine kinase proteins as a potential part of the inflammatory networks of TLR4. The results assumed that MARCKSL1 and creatine kinase proteins might be associated with a statin-induced anti-inflammatory response due to possible interaction with the TLR4.

## 1. Introduction

Macrophages are immune effector cells that perform diverse functions, e.g., innate immune response, inflammatory response, wound healing, etc. [1]. Activation through Toll-like receptor(s) (TLRs) activate macrophages for their function [1,2]. The recognition of pathogenic ligands by TLRs generates signals to downstream adaptor proteins. TLRs lead to activation of transcription factors, e.g., NF-κB, interferon regulatory factors (IRFS), etc., and that induces the expression of type 1 interferon (IFN) different proinflammatory cytokines, e.g., TNF-IL-1, IL-6, IL-12, nitric oxide, reactive oxygen species, etc. [3]. These responses include microbicidal activity, tissue repair, wound healing, inflammation, immune suppression, etc., based on the types and sites of the infection and/or involvements of immune cells and their effectors [1,2,4,5,6]. Apart from innate immune response, Toll-like receptor response in adaptive immunity also plays a role through type 1 interferon production and proliferation of memory T cells [7]. Additionally, unregulated activation of TLRs can lead to autoimmune disease [8]. A comprehensive study of the Toll-like receptor activation signaling network is, thus, essential to find new therapeutic approaches against viral or bacterial infections or autoimmune disorders.

So far, 10 functional Toll-like receptors have been identified in humans. Among TLRs, TLR2 and TLR4 have gained significant attention due to their ability to recognize a diverse array of pathogenic ligands. Toll-like receptor 4, associated with host accessory proteins MD-2 and CD-14, can recognize the LPS from the outer membrane of Gram-negative bacteria [9,10]. Upon activation, TLR4 can exert functions through two distinctive pathways, e.g., MyD88-dependent and TRIF-dependent. Hence, immune signaling through TLR4 is diverse and complicated [11,12]. Thus, we were interested in studying TLR4-interacting immune signaling networks comprehensively.

LPSs are found on the outer membranes of Gram-negative bacteria. During infection, processed LPS can be transferred to the TLR4 complex with MD-2 via LPS-binding protein and CD14. Once bound to the TLR4-MD-2 complex, TLR-4-mediated signaling is activated with the overwhelming secretion of cytokines due to immune inflammatory or host defense response from the immune cells [9,13]. Statins are inhibitors of 3-hydroxy-3-methylglutaryl-CoA (HMG-CoA) reductase. Hyperlipidemic patients are treated with statins to reduce serum cholesterol. Apart from this, statins have been reported with additional immunomodulatory activities. For example, human monocyte-derived dendritic cells’ maturation is suppressed with statins [14].

A few quantitative proteomics studies have been reported with a combined exposure of LPS and statin to immune cells. A label-free proteomics study on LPS-treated human monocyte suggested its potential therapeutic application in tumor treatment [15]. A total of 11 differentially expressed proteins were identified when LPS-treated RAW macrophage cells were analyzed in 2D gel electrophoresis followed by mass spectrometry analysis of peptides [16]. Stable isotope labeling of amino acid in cell culture (SILAC) experiment performed on nuclear and cytosolic fractions of LPS-stimulated macrophages led to the modulation of several mitogen-activated protein kinases (MAPK) and NF-kB signaling pathways [17]. There are reports of quantitative proteomics experiments on isolated lipid rafts from LPS-treated RAW 264.7 macrophage cell line [18,19,20]. A proteomics study on an isolated raft of ABCA1-deficient primary mouse macrophage cells upon LPS stimulation identified 383 unique proteins [20]. In LPS and interferon-γ (IFNγ)-activated macrophages, 409 microtubule-associated proteins were identified [21]. Proteomics analysis of Salmonella-infected macrophage cells identified 244 significantly altered proteins in a time-dependent manner [22]. Isotope-coded affinity tagging (ICAT) profiling of the Raw 264.7 macrophage cells identified 36 differentially expressed proteins upon LPS stimulation [23].

Co-immunoprecipitation (Co-IP) is an established method for studying interacting proteins. In Co-IP, antibody-dependent enrichment of a target protein facilitates the enrichment of interacting proteins [24]. The immunoprecipitated proteins are then analyzed in a Western blot or mass spectrometry based on the project need. The most commonly used proteomics experiment is mass spectrometry-based bottom-up proteomics. In bottom-up proteomics, the proteins are first tryptic digested in small peptides, using proteolytic enzymes, e.g., trypsin. Different proteolytic enzymes can have various preferential sites for proteolytic cleavage. Tryptic digested peptides are separated in liquid chromatography, followed by detection in a mass spectrometer. Application of bottom-up proteomics includes relative and absolute quantification of proteins and peptides, structural analysis, e.g., sequence variation identification, different types of post-translational modification, e.g., PTMs identification, sequence polymorphisms or mutation identification, etc. [25]. All the variations of a gene product present in a sample, e.g., isoforms, different PTMs, single amino acid polymorphism, etc., give rise to different proteoforms [26]. Coexistence of PTMs and sequence variation may be lost in digested peptide in bottom-up proteomics. As proteins are quantified based on the most abundant peptide, proreform-level changes are often lost in bottom-up proteomics [27].

However, Co-IP also has its limitations. Weak protein interaction may be lost during cell lysis [28,29]. Interactions specific to large protein complexes requiring a natural environment may also be lost before detection [30,31]. Chemical cross-linking is another method to study protein–protein interactions. Chemical cross-linking may covalently link and stabilize protein–protein interactions in a natural setting of biological systems. Due to the complexity involved in the analysis of cross-linked modified peptides, for large-scale data sets, it is popular to look for unmodified peptides to search for weak or transiently interacting proteins [32,33,34]. From this perspective, we decided to incorporate our previously published ETD cleavable cross-linker (ETD-XL) in this study [35].

Several proteomics studies were reported on a proinflammatory model system, with LPS being used as a proinflammatory stimulant in RAW 264.7 macrophage cell line. However, no reports of a combined experiment with an anti-inflammatory stimulant, statin, and proinflammatory stimulant LPS have been used in tandem to see the transition in the proteomic network to understand a model system better. Recently, we published a comprehensive study on a TLR2 interactome network using statin and lipopeptide exposures. As far as we know, this is the first study using Co-IP cross-linking proteomics with ETD cross-linkers [36].

In this study, we performed a combination of Co-IP, cross-linking experiment with mass spectrometry-based identification of protein interactors. We applied this strategy in TLR4-mediated immune response concerning pro- and anti-inflammatory stimulus by LPS and statin. We identified 712 proteins through TLR4 pull-down across treated and untreated samples and performed protein network analysis using Ingenuity Pathway Analysis (IPA) software (QIAGEN Inc., https://digitalinsights.qiagen.com/IPA, accessed on 1 May 2018) [37].

## 2. Materials and Methods

### 2.1. Cell Culture, Maintenance, and Sample Preparation

HA-tagged human TLR4 gene transfected stable HEK-293 cell line was purchased from Invivogen (Catalog # 293-htlr4ha). We maintained the hemagglutinin (HA)-tagged TLR4 human embryonic kidney (HEK) 293 cells in Dulbecco’s modified Eagle medium (DMEM with) added 10% fetal bovine serum, 1% penicillin/streptomycin, and antibiotics (50 µg/mL hygromycin and 10 µg/mL blasticidin) in a humidified condition of 5% CO_2_ at 37 °C. Eight experiment sets were prepared with three replicates in each of them.

We treated the cells with 10 µM simvastatin (Sigma-Aldrich now Merck & Co., Inc., Kenilworth, NJ, USA) for 24 h, then stimulated them with 1 µg/mL lipopolysaccharides (LPS-EB, InvivoGen, San Diego, CA, USA) for 1 h in the freshly supplied medium. After collecting the cells, we treated the cells with our in-house ETD cross-linker (XL, 1 µmol/mL) for 30 min, followed by stopping the XL reaction with 50 mM Tris-HCl, pH 8.0. Similarly, we treated cells with simvastatin for 24 h or lipopolysaccharides for 1 h, followed by treatment with an ETD cross-linker. Moreover, we prepared control cell lines with or without the treatment of ETD cross-linker. Then, we performed IP pull-down for proteomics studies, as before [38]. Briefly, cells were lysed with IP lysis buffer containing a protease inhibitor, followed by sonication, incubation, and centrifugation at 20,000× *g*.

### 2.2. Co-Immunoprecipitation (Co-IP) of the TLR4-Interacting Proteins

We used immunoprecipitation, as described before, to separate TLR4-interacting proteins [38]. Briefly, we washed anti-HA magnetic beads with a vortex in TBST buffer and collected supernatant with magnetic beads, followed by overnight rotation at 4 °C. We washed the collected beads with ultrapure water and eluted them in Lamelli buffer. We then performed methanol–chloroform-based precipitation of proteins and trypsin digestion of the recovered proteins. We used label-free spectral counting for the quantification of proteins [36].

### 2.3. In Solution Digestion, Mass Analysis (Nano-LC-MS/MS), and Database Search

In solution, digestion of the immunoprecipitated proteins was performed as described earlier [36]. Briefly, proteins were reduced, alkylated, and then digested overnight with MS-grade trypsin (Promega, Madison, WI, USA) at 37 °C. Formic acid was added to drop pH < 3 and trypsin activity. The samples were then desalted using a C18 desalting column (Thermo Scientific, Rockford, IL, USA). Samples were dried in a speed vacuum and dissolved in 0.1% formic acid, followed by 30-min centrifugation at 20,000× *g*, and analyzed by nano-LC (Ultimate 3000 UHPLC)-MS/MS (Velos Pro Dual-Pressure Linear Ion Trap Mass Spectrometer; ThermoFisher Scientific, Waltham, MA, USA). LC-MS/MS conditions and parameters were followed as before [38]. Briefly, peptides were loaded to a C18 column and ran in a multi-step gradient over 90 min; the MS method was set to carry out an MS2 fragmentation for the top 3 most intense ions. The injection volume was a maximum of 5 µL. Identified proteins were relatively quantified using peptide spectra matches (PSMs) [38].

Proteome Discoverer software (v2.1, Thermo Fisher Scientific, Waltham, MA, USA) and UniProt Human protein database were used to search and match detected spectra to the database and identify the proteins in our samples [39]. Details of the process are described in earlier publications [38]. Briefly, Proteome Discoverer software (v2.1, Thermo Fisher Scientific) was used to search and match our raw files to the database and identify the proteins present in our samples. The reviewed protein sequences of human (Homo sapiens, 120,672 sequences and 44,548,111 residues) were downloaded from the UniProt protein database (www.uniprot.org, accessed on 12 August 2016). The considerations in SEQUEST searches for standard peptides were used, with carbamidomethylation of cysteine as the static modification and oxidation of methionine as the dynamic modification. Trypsin was indicated as the proteolytic enzyme with two missed cleavages. Peptide and fragment mass tolerance were set at ±1.6 and 0.6 Da; a precursor mass range of 350–3500 Da and peptide charges were set, excluding +1 charge state. SEQUEST results were filtered with the target PSM validator to improve the sensitivity and accuracy of the peptide identification. Using a decoy search strategy, target false discovery rates for peptide identification of all searches were <1%, with at least two peptides per protein, a maximum of two missed cleavage. The results were strictly filtered by ΔCn (<0.01), Xcorr (≥measured 1.5) for peptides, and peptide spectral matches (PSMs) with high confidence, that is, with a q-value of ≤0.05. Protein quantifications were conducted using the total spectrum count of identified proteins. Additional criteria were applied to increase confidence that PSMs must be present in all three biological replicates samples. The normalization of identified PSMs among LC-MS/MS runs was carried out by dividing individual PSMs of proteins by total PSMs, and the average % PSM count was utilized for calculating fold changes for different treatment conditions.

### 2.4. Gene Ontology and Protein Interaction Analysis

We functionally categorized the protein-encoding genes using gene ontology systems by the PANTHER classification system corresponding to molecular function, biological process, and cellular components [40]. We generated the heatmap to visualize protein abundances by MeV software (https://sourceforge.net/projects/mev-tm4/, accessed on 2 July 2019) [41]. Differentially expressed proteins’ data were used to create volcano plots using GraphPad Prism v9.3.1 for Windows, GraphPad Software, San Diego, CA, USA, www.graphpad.com (accessed on 11 July 2022). Principal component analysis (PCA) was performed using R package v.4.2.1 (R Core Team (2022), Vienna, Austria) (https://www.r-project.org/, accessed on 1 July 2022) with FactoMineR (v2.4) and Factoextra (v 1.0.7.999) packages, as described previously [42,43,44,45]. We used the protein expression and fold changes with UniProt identifiers to generate core analysis through Ingenuity Pathway Analysis (IPA) (Ingenuity Systems, Redwood City, CA, USA). TLR4 protein interaction networks according to molecular and cellular functions were generated using the Ingenuity Knowledge Base database. The indirect and direct relationships were developed between proteins based on experimentally observed data. Other data sources were considered in the Ingenuity Knowledge Base in humans to create the core analysis [46].

### 2.5. Immunocytochemistry

We conducted immunocytochemistry of control and treated cells as mentioned before [38]. We grew cells on HCl (1 M)-treated glass slides. We then fixed the cells on the slides with 4% paraformaldehyde for 10 min at room temperature. With 0.1% Triton X-100 in 1× PBS, we then permeabilized the cells. Next, we washed the cells with PBS and stained them with Alexa Fluor 488@ phalloidin 2 µL/2 mL in each well for 25 min at RT. Subsequently, we washed the cells with PBS, stained them with propidium iodide (PI) for 5 min, and washed them in PBS before being fixed into coverslips. The cells were then visualized with a Leica DMi8 confocal microscope (Leica, Richmond, IL, USA). The images were examined and analyzed using Las X software (Leica, Richmond, IL, USA).

### 2.6. Statistical Analysis

We used a built-in statistical package in Proteome Discoverer (v2.1) for quantitative analysis of proteins as PSMs. Statistically significant results with *q* ≤ 0.05 were considered for analysis. We used the R package (v3.5.3) for generating scatter plots and a pairwise correlation matrix. The results here were only considered if the correlation coefficient (*R*^2^) was >0.80 (*n* = 3). The codes are provided in https://github.com/mailshahin/ScatterPlotMatrix/projects?type=classic (accessed on 11 July 2022).

## 3. Results

### 3.1. Identification of TLR-4 Interacting Proteins

We wanted to understand the immune responsive interactome in TLR4 mediating immune signaling. We started with Co-IP proteomics on HA-TLR4-HEK293 cells under four conditions: control; LPS; statin; and LPS–statin, with or without post-treatment ETD cross-linker (ETD-XL) (Figure 1). After pull-down with anti-HA magnetic beads, the targeted protein-bound beads were washed and dissolved in Laemelli buffer. After methanol–chloroform purification and reconstitution in trypsin (50 mM ammonium bicarbonate) for in-solution digestion and after in-solution digestion with trypsin, the peptides were analyzed and the data were acquired by nano-LC-MS/MS. The acquired spectra were searched in the UniProt protein database. Peptide spectrum matches (PSMs) were used for the quantification of proteins. Pairwise correlation coefficients among the three biological replicates showed a significant correlation, with an R^2^ value of >0.80 (Appendix A).

A total of 712 proteins were identified and quantified in all four conditions, with or without cross-linker presence. The data set was filtered with at least one unique peptide per protein and a false discovery rate of 1%. Details about the identified proteins and peptides are shown in Appendix A. A total of 416 proteins were identified across four conditions without a cross-linker treatment, whereas 158 were commonly identified in the control; LPS; statin; and LPS–statin without cross-linker treatment. In contrast, 166 proteins were exclusively identified, among which 147 proteins were in the control, 10 in LPS, 6 in statin, and 3 in LPS–statin (Figure 2A). Additionally, we identified 446 proteins in the control and LPS with cross-linker treatment, whereas 165 proteins were commonly identified and 10 proteins were exclusively identified in LPS-treated samples in the presence of cross-linkers (Figure 2B). Similarly, when statin or LPS–statin were compared with cross-linker-treated and untreated control, we identified 12 or 10 proteins exclusively in statin or LPS–statin treated with cross-linker, respectively (Figure 2C,D). Additionally, we identified 75 different proteoforms across all treatments; 16 were uniquely identified without cross-linker treatment and 12 were uniquely identified with cross-linker. (Appendix A). The relative expressions of 712 TLR4-interacting proteins were used to visualize as a heatmap (Appendix A). Proteins with differential expression were plotted to generate volcano plots across control vs. treatment (Figure 3A). The volcano plots show a significant number of differentially expressed proteins (*p* value > 0.05). Principal component analysis (PCA) showed that treatment of cross-linker places each treatment sample in a distant cluster and is well discriminated among the treatments (Figure 3B). Additionally, the heatmap represents that identified proteins were expressed and distributed distinctly upon the treatment of LPS, statin, and LPS–statin in the TLR4 cells (Figure 3C). The 416 proteins identified in the absence of cross-linker in the treatment of statin, LPS, and LPS–statin, were categorized into various gene ontologies using the panther classification system, such as cellular components (six pathways), biological process (ten pathways), and molecular functions (seven pathways) [38]. The ontology pathways showed differential representation across different treatment conditions (Figure 4).

### 3.2. IPA-Based TLR4-Targeted Protein Interactions Network

Ingenuity Pathway Analysis (IPA) was used to perform a core analysis of identified proteins, canonical pathways, hypothetical interaction networks, functional putative upstream regulators, and disease pathways among the TLR4 interactome proteins [38]. This analysis found 14 protein-interacting networks according to top disease and/or functions in each LPS-, statin-, or LPS–statin-treated proteins dataset. TLR4 protein-interacting network was the fourth ranked protein interaction network (Appendix A). The TLR4 network is centered on the cell-to-cell signaling interaction and signaling in all three-stimulus conditions. With LPS, statin, and both statin and LPS stimuli, the TLR4 interacting network is centered on various cancer and organismal injury and abnormalities pathways on LPS, and both statin and LPS responsive samples. With statin stimulus, the TLR4 interacting network is also centered on DNA replication, recombination, repair, and RNA post-transcriptional modification (Appendix A). In this study, we targeted pull down of the TLR4 proteins and their interacting partners. Then, we performed a TLR4-based network analysis (Figure 5, Appendix A) using IPA. We found 35 interacting protein partners using an IPA knowledge-based protein database here. TLR4 protein expression was decreased in statin treatment (Figure 5) and increased with LPS, and both statin and LPS in LPS-stimulated cells. (Appendix A). TLR4 interacted directly with IgG and indirectly with eight different proteins in this putative network. These include IFN-β, interferon-α, actin α-1 (ACTA1), phosphoinositide 3-kinase (PI3K complex), T-cell receptor (TCR), p38 mitogen-activated protein kinases (P38-MAPK), c-Jun N-terminal kinase (JNK), and extracellular signal-regulated kinase (ERK1). The identified four protein kinases interact with cellular kinase-like cyclin-dependent kinase 1 (CDK1) or transcription factors, such as E2F, receptor proteins, such as breast cancer anti-estrogen resistance protein 1 (BCAR1), filamentous proteins, such as ACTA1, filamin-A (FLNA), vimentin (VIM), and F-actin-capping protein subunit α-1 (CAPZA1). This pull-down study identified TLR4 interacting partners, such as kinase CDK1, and filamentous proteins, such as ACTA1, filamin-A, vimentin, and F-actin-capping protein subunit α-1 (CAPZA1), that have various cellular functions in the cell systems. Additionally, we observed that the expression of filamentous proteins increased in LPS treatment upon activation of TLR4 signaling and LPS–statin treatment, whereas it decreased with statin treatment. This phenomenon of filamentous proteins was further confirmed through fluorescence staining, where actin filaments and nucleus were stained.

### 3.3. Protein Identification and Interactions after Cross-Linking Study and Validation

In this study, we used an in-house ETD cleavable cross-linker that covalently captures the low abundance or weakly and transiently interacting proteins in the TLR4 interactome. After cross-linker treatment, we identified 317, 274, 198, and 175 proteins as single peptides, 244, 214, 120, and 116 proteins as two or more peptides in the control, LPS, statin, and LPS–statin as one peptide per protein, respectively (Appendix A). After stringent filtering among cross-linked and non-cross-linked samples, we identified 10, 12, and 10 proteins exclusively in LPS-, statin-, and LPS–statin-treated samples, respectively (Figure 2B,C; Appendix A).

Alexa Fluor 488^®^ phalloidin, a high-affinity filamentous actin probe, was used for F-actin’s selective staining to validate filamentous actin proteins. Propidium iodide was used to bind the DNA in the nucleus (Figure 6). These staining outcomes coincided with previously mentioned findings (Figure 6). From fluorescence staining, it is evident that treatment of LPS increases filamentous protein production and increases cell and nuclear size to activate the cells for immune response. So, the model expression system validated the F-actin protein expression, activation, and suppression of immune response through TLR4 signaling interactome.

## 4. Discussion

In this study, we used HA-TLR4 transfected HEK293 cells as an immune signaling model system to study the TLR4-mediated immune interactome through the atypically expressed HA-TLR4 on a transfected HEK293 cell line. HEK cells are easy to maintain and easy to transfect through foreign DNA. Hence, HEK cells are a popular choice for heterologous proteins’ expression and their functional characterization [47,48,49]. Additionally, due to the suboptimal performance of IP antibodies against TLRs, we had to consider a stable epitope (HA)-tagged TLR4-expressing cell line for our TLR4 immune interactome study design. Moreover, the HEK 293 cells do not have any native TLRs expressed on their membrane, yet they produce downstream functional signaling molecules, proinflammatory cytokines, after TLR4 ligand stimulation [38,50]. This makes them an ideal system to control the expression of a selected TLR without endogenous background receptors. Hence, we chose HA-TLR4 transfected HEK293 cells for targeted immune precipitation against HA-tagged TLR4 and cross-linking proteomics experiment to study TLR4 mediated immune signaling.

In this study, we used a Co-IP-based mass spectrometry approach along with an ETD cross-linker that helps to identify low abundance proteins. This approach pulled down TLR4-bound proteins to reveal the TLR4 interacting partners associated with immune response. However, Co-IP-based approaches enrich a significant number of nonspecific proteins. In addition, a low-resolution mass spectrometry platform was used, which could limit the total protein coverage. Fortunately, the covalent labeling of proteins with cross-linkers helps to reduce nonspecific proteins significantly.

We have used two antagonistic stimulants, e.g., LPS, a proinflammatory stimulant [17], and statin, an anti-inflammatory stimulant [51,52], and, in tandem, the first statin then LPS to realize a bigger picture, where we can evaluate immune signaling in proinflammatory and in anti-inflammatory condition, and also its transition from anti-inflammatory to proinflammatory condition. The introduction of a cross-linker into the study design enabled us to capture and enrich low-abundance, transiently interacting protein partners during these three different above-mentioned immune signaling conditions.

After immune precipitation, tryptic digestion, LC-MS/MS analysis, and database search, we were able to obtain lists of proteins expressed in mentioned conditions. We used the PSM value to calculate normalized PSM percentage and fold changes among these conditions compared to the control sample. From this data set, we used the gene accession number lists (Appendix A) of specific treatment conditions to generate a Venn diagram and identified the proteins that are exclusively expressed in each specific condition (Figure 2, Appendix A). After stringent filtering, we identified 10, 12, and 9 proteins exclusively in the cross-linked samples treated with LPS, statin, and both statin and LPS together. The normalized PSM percentage values from different conditions were used to create a heat map (Figure 3, Appendix A) showing each protein’s differential expression across all treatment conditions. The gene accession number lists were also used to generate gene ontology information of identified proteins across all treatment conditions and the proteins were classified according to cellular components, biological processes, and molecular functions. All ontology conditions showed differential expression across the three categories and each subcategory (Figure 4, Appendix A). The fold change data (Appendix A) we calculated from PSMs were used to generate core analysis through Ingenuity Pathway Analysis. Hypothetical interaction networks, canonical pathways, and functional and disease pathways were constructed, and putative upstream regulators of TLR4 interactome were identified (Figure 5, Appendix A). In this analysis, TLR4 showed LPS-dependent higher expression. The network showed protein interactions directly and indirectly to TLR4, and, at the same time, their comparative expressions. Noteworthy interacting partners included different cytokines (e.g., IFN-α and IFN-β), different kinases (e.g., p38 mitogen-activated protein kinases, c-Jun N-terminal kinase, extracellular signal-regulated kinase, etc.), and different filamentous proteins (e.g., actin α-1, filamin-A, vimentin, and F-actin-capping protein subunit α-1).

The appearances of different categories of proteins, receptors, transcription factors, kinases, and filamentous proteins in the network suggested the possibility of complex connections among proteins in the proposed TLR4 interactome (Figure 5, Appendix A). In this study, we also observed that the filamentous protein (e.g., vimentin, F-actin, and filamin) expressions go up with LPS stimulation, go down with statin stimulation, and go up again compared to statin once LPS is reintroduced in tandem after removal of statin (Figure 5, Appendix A). This corresponds to similarly published literature, where actin, filamin A, or vimentin expressions were reported to be directly related to macrophage activation and function [53,54,55]. This trend was further confirmed through the fluorescence staining experiment where Alexa Flour 488@ phalloidin selectively probed F-actin and propidium iodine bound the DNA in the HEK293 nucleus (Figure 6). We observed an increase in filamentous protein production, and cell and nuclear size upon LPS stimulation. So, our model system with atypically expressed HA-TLR4 in HEK293 cells showed similar immune responsive gene expression patterns as regular macrophages and, thus, is a good model system to further explore a novel responsive protein in the TLR4 interactome. As mentioned earlier, incorporating the cross-linking step in the experiment enriched low-abundance and transiently interacting proteins in the TLR4 interactome. So, from the protein lists, which were exclusively identified in the cross-linked samples (Figure 2, Appendix A) along with the treatment of LPS, statin, and LPS-statin, we wanted to choose candidate proteins. Creatine kinase is a marker of kidney function, and [56] its level can be elevated due to trauma and muscle injury [57]. This enzyme catalyzes the reversible transfer of a γ-phosphate group of ATP to the guanidino group of creatine to yield phosphocreatine (PCR). In skeletal muscle, a large pool of phosphocreatine is used for ATP regeneration [58]. As we have observed the presence of different kinases in our IPA-generated TLR4 network, creatine kinase appeared in our lists of exclusively identified transient/weak interacting proteins in the presence of a cross-linker. We hypothesize that creatine kinase may also have some role in TLR4 signaling, which may or may not be for ATP regeneration. Myristoylated alanine-rich C kinase substrate (MARCKS) and MARCKSL1 proteins are protein kinase C (PKC) substrates that participate in myriad functions in the living system. Both share identical effector domains, binding to calmodulin in a phosphorylation-dependent manner [59]. MARCKS have been implicated with membrane–cytoskeletal signaling, integrin activation, cell spreading, cell–cell adhesion, migration, and phagocytosis [60,61,62].

MARCKS had been reported to be expressed in macrophages through LPS stimulation: in Madin–Darby canine kidney (MDCK) epithelial cells and renal tubule cells, MARCKS is endogenously expressed [63,64]. Previously, it showed that treatment with IFN-γ and TNF-α in epithelial cells increased MacMARCKS (MRP/MARCKSL1) expression. Treatment of statin ensures direct suppression of cytokines (e.g., IFN-gamma, tumor necrosis factor (TNF)-α, interleukin (IL)-2, and IL-4 [65]). Baicalein pretreatment associated with cytokines’ (e.g., IL-6 and TNF-α) suppression was reported along with suppression of creatine kinase concentration [66]. We have observed high creatine kinase expression in cross-linker-treated, and LPS-stimulated samples and creatine kinase brain isoform for cross-linker treatment with a statin. We also observed differential expression of MARCKS-related protein in the combined treatment of statin and cross-linker and statin–LPS and cross-linker (Appendix A). Thus, we can hypothesize that both MARCKS and creatine kinase expressions may be critical for inducing cytokines and play a role in TLR-4 mediated signaling pathways.

Co-IP cross-linking proteomics study is now becoming a very popular technique due to its ability to pull down transient and weak interactors along with stable ones. In addition, employing strong denaturing washing conditions further facilitates the removal of nonspecific interactors. Although a good approach, it suffers from some potential drawbacks. Sometimes it is difficult to pinpoint the direct and indirect interactions if interactors were identified using non-cross-linked peptides. An efficient and in-depth bioinformatics approach will overcome these limitations in the future to identify cross-linked peptides with high confidence in large-scale experiments. With the bottom-up proteomics approach combined with only trypsin, we cannot fully annotate the proteoform expression across different conditions. Proteoform consists of alternative splice variants, e.g., isoform, post-translational modifications, and coding single-nucleotide polymorphisms [67]. This study could only identify certain isoforms of proteins. In this case, detailed quantitative identification of proteoforms requires a combination of mass spectrometry intensive top-down proteomics combined with the use of multiple types of proteases in a bottom-up-type approach [27]. We want to acknowledge that our proteomics method identified two potential biomarkers, but further validations with alternative approaches, including the cross-linked peptide identifications, are necessary to designate them as bona fide interactors in statin induced TLR4 signaling pathways. Nevertheless, our method pinpointed these two interactors for further explorations.

## 5. Conclusions

An immunoprecipitation-based chemical cross-linking proteomics approach was implemented in an HA-TLR4 HEK293T cell line to decipher the interactome of TLR4 with the treatment of pathogenic ligand and drug. For these studies, we utilized LPS, statin, and both statin and LPS to understand the effect of pro- and anti-inflammatory responses on TLR4 signaling pathways. Additionally, we used a compact ETD cleavable chemical cross-linker to capture weak and transient interactions with TLR4 protein. The immunoprecipitated and cross-linked proteins were digested on beads, and the peptides were analyzed through high-throughput liquid chromatography–mass spectrometry (LC-MS/MS). The label-free quantitation technique using PSMs measured the proteins’ relative expression between treated and untreated samples with or without a cross-linker. We identified a total of 712 proteins altogether and generated TLR4-targeted protein networks using Ingenuity Pathway Analysis (IPA) software. The outcomes suggested MacMARCKS and creatinine protein potential involvement with a statin-induced anti-inflammatory response due to possible interaction with the TLR4. Those candidate proteins need to be further validated and evaluated concerning immune responses. That will help to understand the molecular processes of TLR4-mediated protein interactions.

## Figures and Tables

**Figure 1 proteomes-10-00031-f001:**
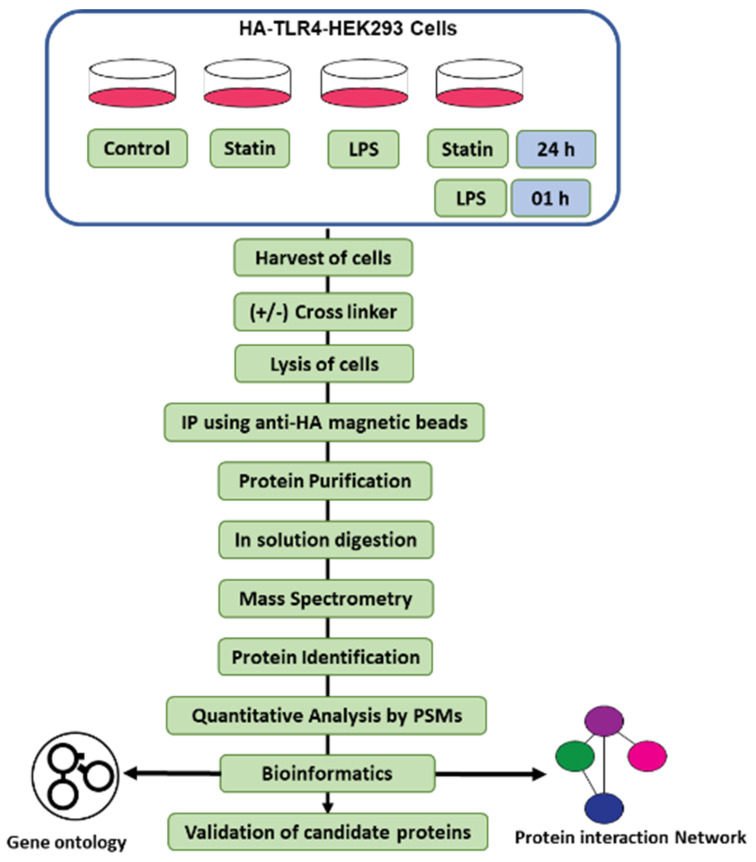
Experimental overview of the IP-cross-linked-based proteomics analysis. HA-TLR4-HEK293 cells were treated with a statin, LPS, and alone or in tandem with or without cross-linkers as described. Pull-down samples were digested in solution with trypsin and analyzed by nano-LC-MS/MS, followed by PSM-based quantitative analysis. All proteomics studies have been carried out in triplicate.

**Figure 2 proteomes-10-00031-f002:**
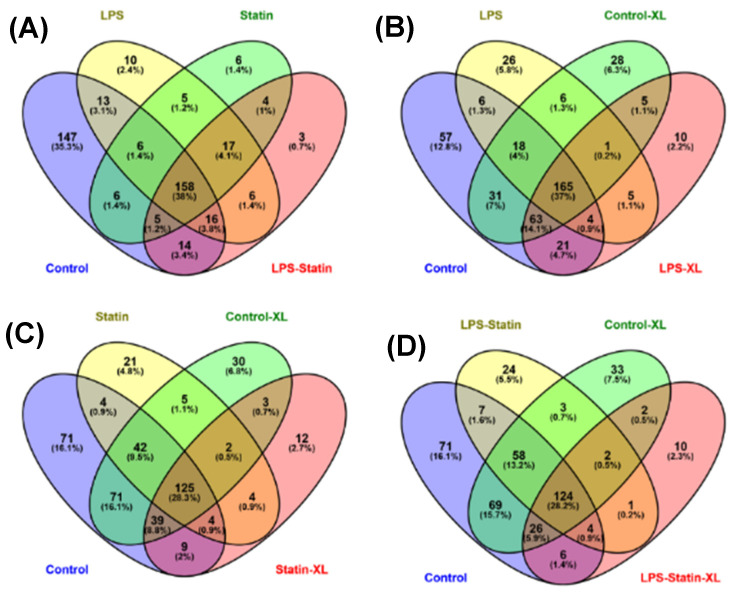
Venn diagrams represent the allocations of identified proteins across all treatment conditions. For example, the diagram shows the distribution of the total and exclusively identified proteins in HEK293 cells upon treatment with (**A**) statin, LPS, and LPS–statin; (**B**) control cross-linker and LPS cross-linker; (**C**) control cross-linker and statin cross-linker; and (**D**) control cross-linker and LPS–statin cross-linker.

**Figure 3 proteomes-10-00031-f003:**
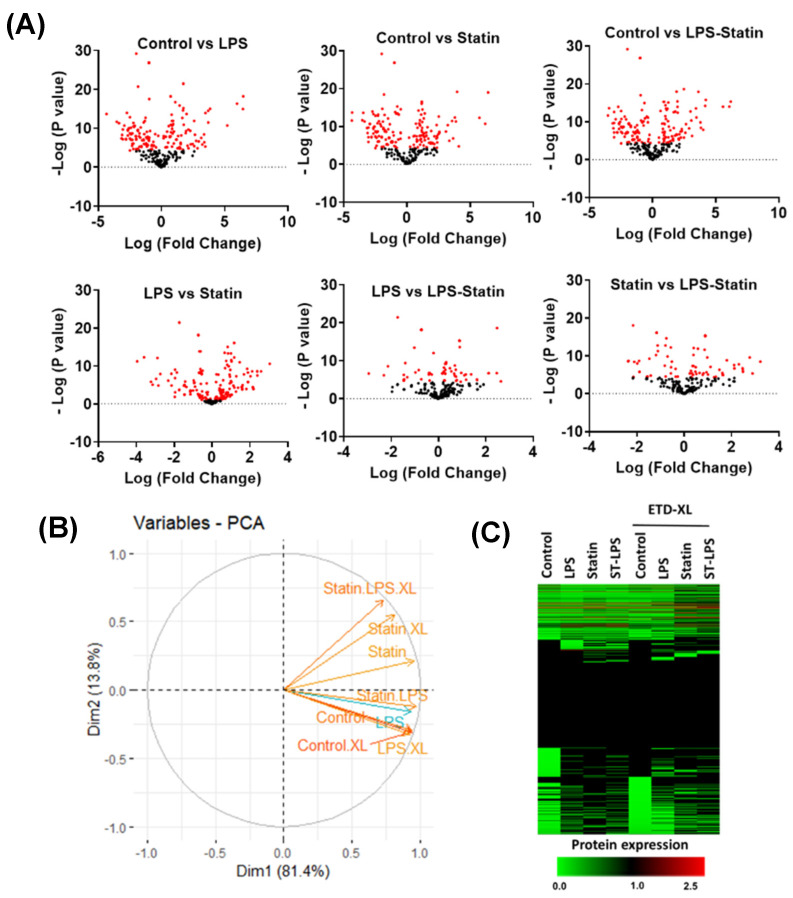
Alternation of proteins upon treating LPS and statin in HA-TLR4-HEK293 cells. (**A**) Volcano plots show relative expression of identified proteins under four conditions: control, LPS, statin, and LPS–statin. Proteins were differentially altered across the different treatment conditions, e.g., statin, LPS, and both statin and LPS. (**B**) Principal component analysis (PCA) reveals that cross-linker treatment shows distinct discrimination among treated samples. (**C**) Heatmap showing relative expression across control and treatment samples with or without cross-linkers.

**Figure 4 proteomes-10-00031-f004:**
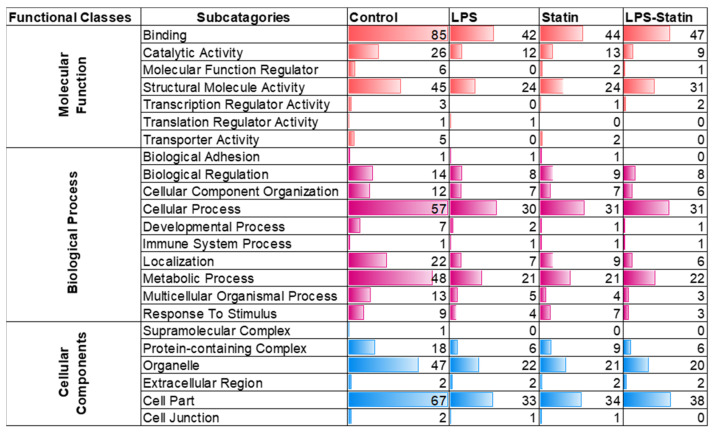
Analysis of gene ontology enrichment analysis. Gene ontology classes were enriched based on the identified proteins’ molecular functions, biological processes, and cellular components.

**Figure 5 proteomes-10-00031-f005:**
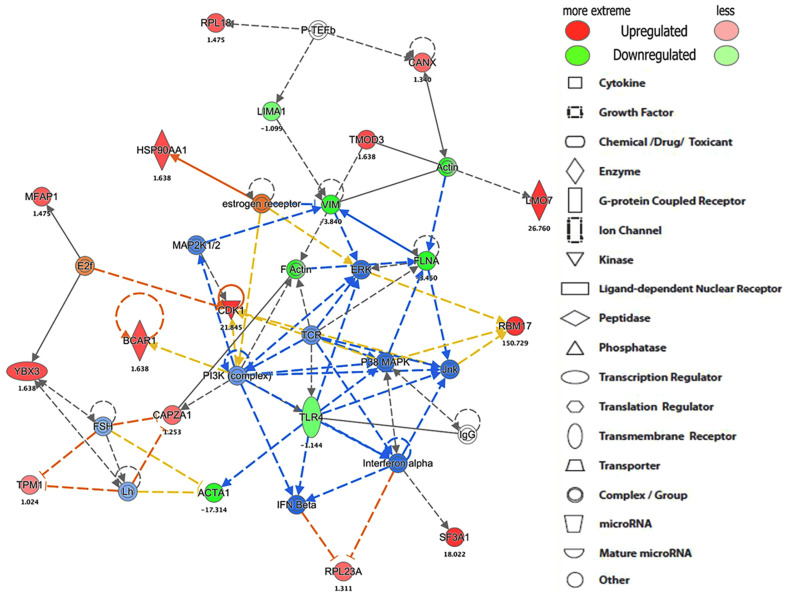
The protein networks show TLR4-interacted proteins and their interacting partners with expression. The representative protein interaction network was generated using IPA software upon statin treatment in HA-TLR4-HEK293 cells.

**Figure 6 proteomes-10-00031-f006:**
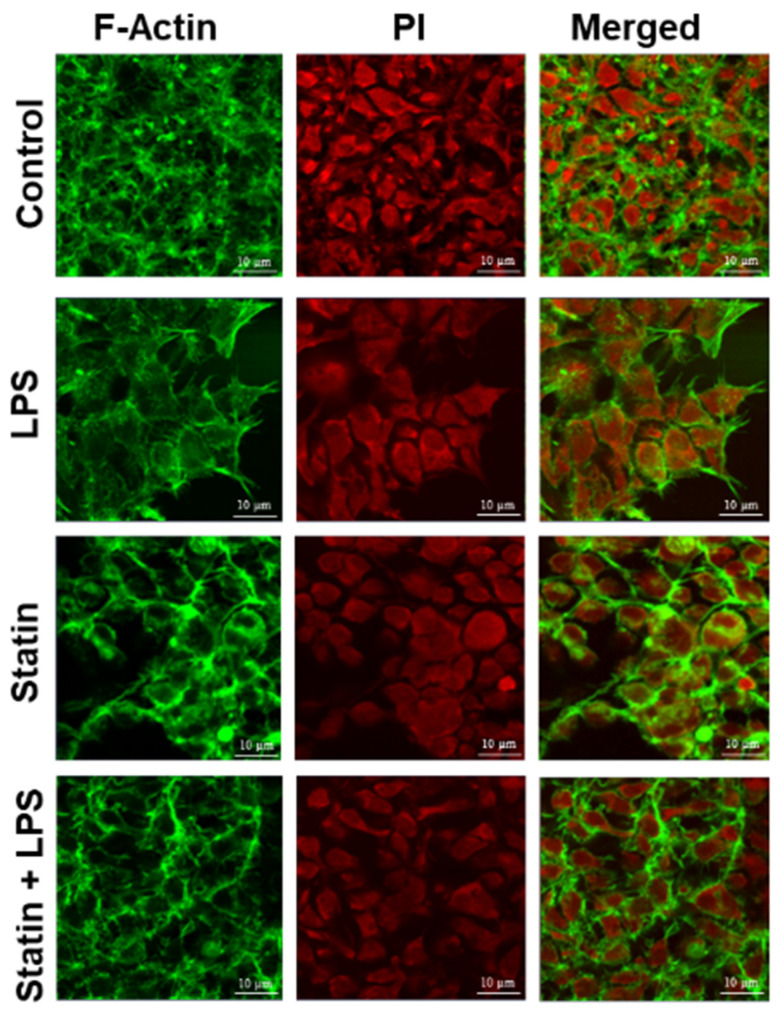
Validation of putative marker proteins. Fluorescence staining of F-actin and nucleus. HA-TLR4-HEK293 cells were stained with Alexa 488 conjugated phalloidin probe (green) and propidium iodide (red).

## Data Availability

The mass spectrometry proteomics data have been deposited to the ProteomeXchange Consortium via the PRIDE partner repository with the dataset identifier PXD014661.

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
