# Peer review of "Identification of Inflammatory Proteomics Networks of Toll-like Receptor 4 through Immunoprecipitation-Based Chemical Cross-Linking Proteomics"

_proteomes, 2022, doi:10.3390/proteomes10030031_

Round 1

Reviewer 1 Report

1) Line 101 - Please change sub title appropriately, example - Maintenance of cell culture and sample preparation. 

2) In line 125 Section - Please include database search parameters such as fixed modifications, variable modifications, precursor mass tolerance, database size (number of proteins), FDR, minimum number peptides per protein etc. Although the reference has been added since it is a Proteomics journal it would be helpful to include these details in the text.  

3) Provide details about what R packages were used in the section "Statistical analysis" line 168. 

4) Which statistical test was applied in proteome discoverer 2.1?

5) Please provide link in the for MeV software used for generating heat maps.

6) Fig 1: In the schematic please show petri plates for each parameter with labels using concentration, time of incubation and number of replicates. 

7) Please show volcano plots and PCA plot in fig 3

8) It is not clear if the proteins identified with 1 protein unique peptide also had shared peptides if yes then they should also be included in the excel - as in for identification all shared peptides should be used but for quantification protein unique peptide/s should be used. Please elaborate. 

9) Please add numbers on top of bar chart for each group represented by different colour. 

10) In fig 4 cellular compartment is cell part equivalent to cytoplasm? If possible use CC analysis from IPA. 

11) Please show some quantitation and apply statistics for immunohistochemistry and western blots. 

12) Please provide password for ProteomeXchange repository so the raw data can be checked. 

Author Response

Reviewer 1

1) Line 101 - Please change sub title appropriately, example - Maintenance of cell culture and sample preparation. 

Response: Thank you so much. We edited the subtitle.

2) In line 125 Section - Please include database search parameters such as fixed modifications, variable modifications, precursor mass tolerance, database size (number of proteins), FDR, minimum number peptides per protein etc. Although the reference has been added since it is a Proteomics journal it would be helpful to include these details in the text.  

Response: Thank you so much. We updated in methods.

3) Provide details about what R packages were used in the section "Statistical analysis" line 168. 

Response:  Thank you so much. We updated the R version and did not use any additional package. We provided the code link to github as well as in text.

4) Which statistical test was applied in proteome discoverer 2.1?

Response: During search proteome discoverers’ built in search parameter False Discovery rate was set to 0.01 (q-value) as strict filtering so as less than 1 percent chance of random match between the true data base and decoy database.

5) Please provide link in the for MeV software used for generating heat maps.

Response:  Thank you so much. We provided the weblink in text.

6) Fig 1: In the schematic please show petri plates for each parameter with labels using concentration, time of incubation and number of replicates. 

Response: Thank you so much. We updated the Fig. 1.

7) Please show volcano plots and PCA plot in fig 3

Response: Thank you so much. We added and updated the Fig. 3.

8) It is not clear if the proteins identified with 1 protein unique peptide also had shared peptides if yes then they should also be included in the excel - as in for identification all shared peptides should be used but for quantification protein unique peptide/s should be used. Please elaborate. 

Response: This is provided in Supplementary-Table - S1 e.g. at least one peptide or more identified for each proteins. For identification we considered at least one peptide identified. For Relative quantification calculation, at least two peptides identified per protein was considered.

9) Please add numbers on top of bar chart for each group represented by different color. 

Response: Thank you so much. We changed and updated with number of protein encoding genes.

10) In fig 4 cellular compartment is cell part equivalent to cytoplasm? If possible use CC analysis from IPA. 

Response: Unfortunately, IPA is a very expensive software and our current license expired. We could not do the analysis the reviewer requested.  

11) Please show some quantitation and apply statistics for immunohistochemistry and western blots. 

Response: We added and updated.

12) Please provide password for ProteomeXchange repository so the raw data can be checked. 

Response: Data Deposited in Ebi pride server. Details provided here and data availability section.

Website: http://www.ebi.ac.uk/pride

 Project accession: PXD014661

 Project DOI: Not applicable

Reviewer account details:

Username: [email protected]

 Password: dlLGHVUt

Reviewer 2 Report

In this study, HA-labeled TLR4 monoclonal antibody was used for immunoprecipitation to remove the protein interacting with TLR4. Protein networks were also constructed and analyzed. Subsequently, the paper suggested MARCKS protein potential involvement with statin-induced anti-inflammatory response due to possible interaction with the TLR4. To be sure, the overall writing of the paper is good.

This paper mainly has the following deficiencies:

1 Lack of innovation. The novel point of this research is to investigate the effect of statins on TLR4 by immunoprecipitation. The innovation degree is not enough, and the research content is lack of significance.

2 This paper experimented with Co-IP, but does not reflect the characteristics of the system, not compared with other methods of has been put forward at the same time, makes the experiment not convincing enough. In addition, the workload is small and downstream validation experiments are lacking.

3 There are some formatting errors in the paper, such as “CO2” “℃”

4 The method description is incomplete. Western blot was not statistically analyzed. Conditions such as antibody concentration should be described as completely as possible.

5 The figure is not complete, Figure 6. A should be marked with scale and the molecular weight of the protein should be specified.

Author Response

Reviewer 2

In this study, HA-labeled TLR4 monoclonal antibody was used for immunoprecipitation to remove the protein interacting with TLR4. Protein networks were also constructed and analyzed. Subsequently, the paper suggested MARCKS protein potential involvement with statin-induced anti-inflammatory response due to possible interaction with the TLR4. To be sure, the overall writing of the paper is good.

This paper mainly has the following deficiencies:

1 Lack of innovation. The novel point of this research is to investigate the effect of statins on TLR4 by immunoprecipitation. The innovation degree is not enough, and the research content is lack of significance.

Response: AP-MS is widely used for comprehensively studying protein networks.  Combining AP-MS with cross-linking is now becoming popular in mass spectrometry-based proteomics.  This technique not only captures transient and weak interactors but also helps to remove nonspecific interactions very efficiently using denaturing washing solution due to the covalent attachment of the protein to protein.  This is the first time this technique was applied to decipher TLR4 protein networks by stimulating its pathogenic ligand LPS and a widely used drug statin.  A cutting-edge publication using AP-CXL-proteomics was done by our group using TLR2 as bait before and published in no one proteomics journal, "Molecular and Cellular Proteomics".  We hope this statement will clarify the innovation and "significance" concerns of the reviewer.

2 This paper experimented with Co-IP, but does not reflect the characteristics of the system, not compared with other methods of has been put forward at the same time, makes the experiment not convincing enough. In addition, the workload is small and downstream validation experiments are lacking.

Response: The study was an extensive pilot proteomics study to identify the novel interactor of TLR4. We have future plan to extend and validate the identified protein using molecular techniques.

3 There are some formatting errors in the paper, such as “CO2” “℃”

Response: We added and updated.

4 The method description is incomplete. Western blot was not statistically analyzed. Conditions such as antibody concentration should be described as completely as possible.

Response: Thank you so much. We added and updated.

5 The figure is not complete, Figure 6. A should be marked with scale and the molecular weight of the protein should be specified.

Response: Thank you so much. We added and updated. We added the molecular weight and dilution in methods.

Reviewer 3 Report

The manuscript entitled “Identification of inflammatory proteomics networks of toll-like receptor 4 through immunoprecipitation-based chemical cross- 3 linking proteomics” by Shahinuzzaman et al., used CO-IP, crosslinking, and proteomics to investigate the effect of pro and anti-inflammatory responses through TLR4 receptor. The study has the potential to advance the understanding of the subjects, however, the manuscript in the current form is not coherent and does not meet the criteria for publication in journal Proteomes. The results are not presented in a clear manner, and it is not convincing as to why a particular experiment was done and what is the outcome of the experiment, it appears very haphazard in its current form. Some of the major points are listed below-

1.       The manuscript needs to be rewritten with the major English assistance. In many of the paragraphs it is not clear what the author wants to convey, it’s very confusing at times.

2.       In line 27 author writes “We identified 712 differentially expressed proteins in the treated sample compared to untreated samples” however, in line 193 author writes “Overall, 712 proteins were identified and quantified in all four conditions, with or without crosslinker presence”. It is not clear whether 712 is the total number of identified protein or total number of differently expressed protein. Clearly there should be a difference in total number of proteins identified across all condition and number of proteins whose expression (abundance) changes significantly upon treatment under different conditions?

3.       Please elaborate the purpose of crosslinking experiment in the context of the study, what was the aim and whether the aim was achieved as it is not clear from the current writing in the manuscript.

4.       It is not clear how the analysis of crosslinked peptide was done? Please provide details of analysis of cross-linked peptides, what is the mass of the cross linker, which software was used for searching etc.?

5.       Please provide details of the ETD crosslinker or provide reference if it is already published?

6.       In Figure3 it is not clear which proteins are visualized, what was the criteria for selection of these proteins? Was it an unsupervised or supervised clustering?

7.       How many of the previously known interacting partner of TLR4 were identified? In Figure 5, none of the well described interacting partner like TIRAP, TRAF6, TICAM1, IRAk1, HMGB1, LY96, Ly86, HSPd1, CD14 of TLR4 are shown, which raises the question on specificity of pull down and cross-linking experiment.

8.       The results presented are confusing and contradictory, for example, in line 277 author writes “Creatine kinase and MacMARCKS were exclusively observed in the presence cross-linked upon the treatment of statin, and LPS-Statin.”, however, in the validation experiment in Western blot (Figure 6B) Creatine kinase and MacMARCKS bands are present in control as well.

Author Response

Reviewer 3

The manuscript entitled “Identification of inflammatory proteomics networks of toll-like receptor 4 through immunoprecipitation-based chemical cross- 3 linking proteomics” by Shahinuzzaman et al., used CO-IP, crosslinking, and proteomics to investigate the effect of pro and anti-inflammatory responses through TLR4 receptor. The study has the potential to advance the understanding of the subjects, however, the manuscript in the current form is not coherent and does not meet the criteria for publication in journal Proteomes. The results are not presented in a clear manner, and it is not convincing as to why a particular experiment was done and what is the outcome of the experiment, it appears very haphazard in its current form. Some of the major points are listed below-

  1. The manuscript needs to be rewritten with the major English assistance. In many of the paragraphs it is not clear what the author wants to convey, it’s very confusing at times.

Response: Thank you so much for your suggestions. We revisited and edited the manuscript.

  1. In line 27 author writes “We identified 712 differentially expressed proteins in the treated sample compared to untreated samples” however, in line 193 author writes “Overall, 712 proteins were identified and quantified in all four conditions, with or without crosslinker presence”. It is not clear whether 712 is the total number of identified protein or total number of differently expressed protein. Clearly there should be a difference in total number of proteins identified across all condition and number of proteins whose expression (abundance) changes significantly upon treatment under different conditions?

Response: 712 is the total number of proteins identified across all conditions. This will be a pairwise comparison to calculate significant changes across different conditions, as shown in the newly added volcano plot. The significant change compared to control may not be the same as Cross linker treatment. So, to keep things simple, we did not comment on the total number of proteins changed across different conditions. Although significantly changed proteins are available across a pairwise comparison of the volcano plot list, they can be provided in the supplementary data if required. 

  1. Please elaborate the purpose of crosslinking experiment in the context of the study, what was the aim and whether the aim was achieved as it is not clear from the current writing in the manuscript.

Response: We added some more information’s about the crosslinking experiments.

“Chemical cross-linking may covalently link and stabilize protein-protein interactions in a natural setting of biological systems. Due to the complexity involved in the analysis of cross-linked modified peptides, for large-scale data sets, it is a popular approach to look for unmodified peptides to search for interacting proteins with weak or transient interactions [29-31]. From this perspective, we decided to incorporate our previously published ETD cleavable cross-linker (ETD-XL) in this study [32].”

  1. It is not clear how the analysis of crosslinked peptide was done? Please provide details of analysis of cross-linked peptides, what is the mass of the cross linker, which software was used for searching etc.?

            Response:This study used cross-linking to capture transient/weak/stable interactors during pulldown studies. We did not look for the cross-linking sites by analyzing MS/MS of cross-linked peptides. The biggest challenge still exists when we look for large-scale samples due to software performance. Nevertheless, our group is developing new software capabilities to look into more details on the sites of the interactions.

  1. Please provide details of the ETD crosslinker or provide reference if it is already published?

Response: Its already published and cited in text as reference number 24. For convenience link provided here. https://www.sciencedirect.com/science/article/abs/pii/S1874391920302141

  1. In Figure 3 it is not clear which proteins are visualized, what was the criteria for selection of these proteins? Was it an unsupervised or supervised clustering?

Response: The criteria for sections for these proteins are differential protein expressions in the proteomics experiments. The large number of the proteins name is very difficult to show in a heatmap. By the way, supplementary Table 2 has the names of the proteins used in the heatmap. Also, with added volcano plot now, (figure list of significantly changed protein accession numbers can be provided now as supplement. 

  1. How many of the previously known interacting partner of TLR4 were identified? In Figure 5, none of the well described interacting partner like TIRAP, TRAF6, TICAM1, IRAk1, HMGB1, LY96, Ly86, HSPd1, CD14 of TLR4 are shown, which raises the question on specificity of pull down and cross-linking experiment.

Response: Some of the genes pulled as linker genes. We tried to verify our results with existing knowledge-based database through IPA, and correlated the expression upon the LPS, and statin treatments.

  1. The results presented are confusing and contradictory, for example, in line 277 author writes “Creatine kinase and MacMARCKS were exclusively observed in the presence cross-linked upon the treatment of statin, and LPS-Statin.”, however, in the validation experiment in Western blot (Figure 6B) Creatine kinase and MacMARCKS bands are present in control as well.

Response: We understand the reviewer's concerns. Sometimes we see this discrepancy in comparing proteomics and western blot analysis. Discovery-based proteomics identifies targets and further validations are necessary to pinpoint the targets using molecular biology techniques. Here western blot (WB) analysis helped us to focus on MacMARCKES than creative kinase. WB is very sensitive technique for detecting protein expression. Here WB was used for validating the MS results. The results gave exciting information during treating LPS and Statin in TLR4-HEK293 cells.

Round 2

Reviewer 2 Report

I think this manuscript can be received.

Author Response

Thank you so much. We address the editor's comments.